# State-of-the-art IoV trust management a meta-synthesis systematic literature review (SLR)

Abdul Rehman[1], Mohd Fadzil Hassan[1], Kwang Hooi Yew[1], Irving Paputungan[2] and Duc Chung Tran[3]

[1] Centre for Research and Data Science (CeRDaS), Computer and Information Science Department, Universiti Teknologi Malaysia, Seri Iskandar, Perak Darul Ridzuan, Malaysia
[2] Informatics Department, Universitas Islam Indonesia, Daerah Istimewa, Yogyakarta, Indonesia
[3] Computing Fundamental Department, FPT University, Hoa Lac Hi-Tech Park, Hanoi, Vietnam



## ABSTRACT

In the near future, the Internet of Vehicles (IoV) is foreseen to become an inviolable part of smart cities. The integration of vehicular ad hoc networks (VANETs) into the IoV is being driven by the advent of the Internet of Things (IoT) and high-speed communication. However, both the technological and non-technical elements of IoV need to be standardized prior to deployment on the road. This study focuses on trust management (TM) in the IoV/VANETs/ITS (intelligent transport system). Trust has always been important in vehicular networks to ensure safety. A variety of techniques for TM and evaluation have been proposed over the years, yet few comprehensive studies that lay the foundation for the development of a "standard" for TM in IoV have been reported. The motivation behind this study is to examine all the TM models available for vehicular networks to bring together all the techniques from previous studies in this review. The study was carried out using a systematic method in which 31 papers out of 256 research publications were screened. An in-depth analysis of all the TM models was conducted and the strengths and weaknesses of each are highlighted. Considering that solutions based on AI are necessary to meet the requirements of a smart city, our second objective is to analyze the implications of incorporating an AI method based on "context awareness" in a vehicular network. It is evident from mobile ad hoc networks (MANETs) that there is potential for context awareness in ad hoc networks. The findings are expected to contribute significantly to the future formulation of IoVITS standards. In addition, gray areas and open questions for new research dimensions are highlighted.

## INTRODUCTION

After the rise of the Internet of Things (IoT), several new technologies have merged to form vehicular ad hoc networks (VANETs), which have evolved significantly over the last decade. The IoT has attracted researchers' attention in that it enables the unification

Corresponding author
Abdul Rehman,
abdul_18000023@utp.edu.my

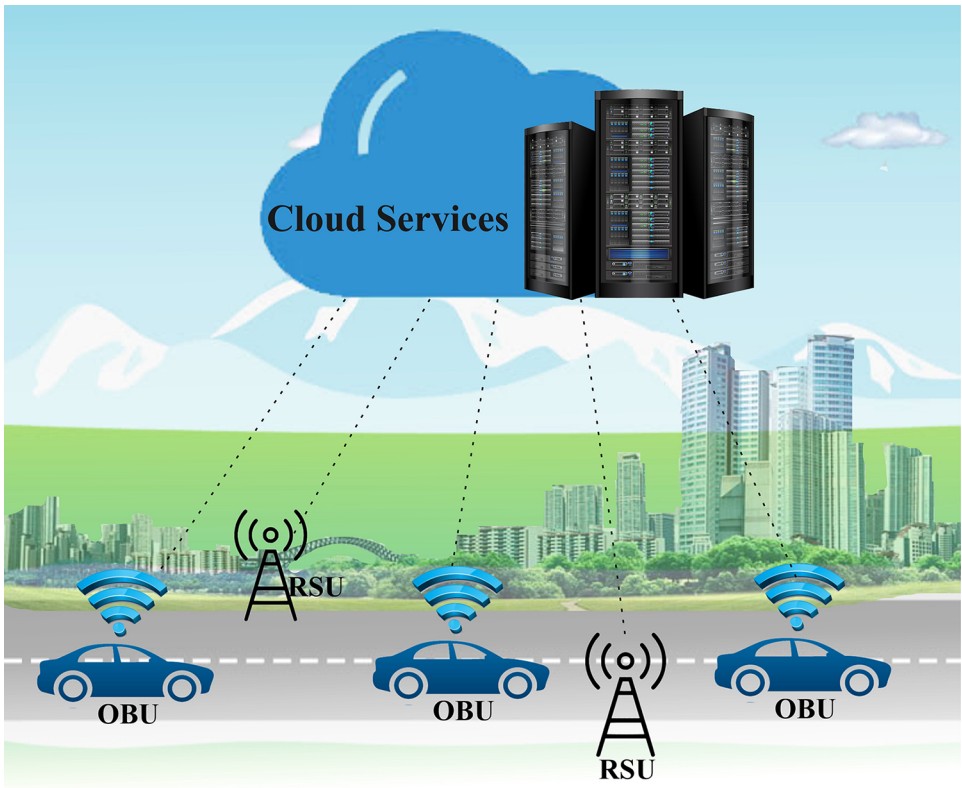

**Figure 1 An overview of the main components of the IoV.**

of VANETs with the Internet (*Alam, Saini & El Saddik, 2015*; *Hamid, Zamzuri & Limbu, 2019*; *Xie et al., 2017*). The IoV and social Internet of Vehicles (SIoV) have set in motion a paradigm shift in vehicle technology and innovation (*Contreras-Castillo, Zeadally & Guerrero-Ibañez, 2018*; *Jabri et al., 2019*; *Zia, Shafi & Farooq, 2020*). An overview of the main components of the IoV, which is presented in Fig. 1, includes vehicles with an on-board unit (OBU), a roadside unit (RSU), and cloud connectivity. VANETs and ITSs are looking forward to the implementation of the IoV (*Alam, Saini & El Saddik, 2015*) as an essential part of smart cities as a future vision of metropolitan areas (*Contreras-Castillo, Zeadally & Guerrero-Ibañez, 2018*; *Fangchun et al., 2014*). Vehicle manufacturers and smart city planners alike are looking forward to equipping their vehicles with smarter technology. Currently, implementation issues such as safety, policy, infrastructure standards, ethics, and specifications still exist, although the next wave of technology aims to implement vehicular networks either partially or fully.

The wireless nature of vehicular networks increases their vulnerability to threats (*Alzamzami & Mahgoub, 2020*; *Hasrouny et al., 2017*). Malicious vehicles could divert traffic according to an organized strategy that may lead to a disaster. A malicious activity would lead to traffic congestion and accidents, resulting in a possible risk to human life. The Internet connectivity to the vehicular network has increased the attack surface. The flexibility of vehicular networks needs to be such that they are able to accommodate

vehicles that join and leave the network at any point in time (*Patel & Jhaveri, 2015*). These networks have no geographical limits because of their mobile characteristics. In this situation, vehicle-to-vehicle (V2V) communication is very significant. The availability of a centralized network is a common problem in VANETs, which means that vehicles may come under an unsupervised network at times. The disadvantage of a centralized network is the well-known single point of failure (SPoF) problem, where only a single point of disruption could lead to network unavailability. Previously (*Fangchun et al., 2014*), a broad overview of the IoV and related future challenges, such as communication, security, and dynamism, was presented. Another study discussed the basic visionary idea of the IoV (*Mbelli & Dwolatzky, 2016*), including potential benefits in coming years, and security was also considered as being an important concern for the future of the IoV. The development of the IoV greatly depends on the availability of intelligent systems for which there currently is a great need (*Fangchun et al., 2014*). VANETs present multidimensional security problems, and make security one of the basic challenges for ad hoc networks. The available trust evaluation and management solutions for VANETs / IoV need to be improved (*Qiu et al., 2016*; *Sumithra & Vadivel, 2018*; *Yao et al., 2017*).

Security, privacy, standards, and many other challenges would have to be overcome before a full-fledged vehicular network could be implemented (*Lin et al., 2008*). Vehicular networks possess unique characteristics in addition to those of many other types of networks, in that they require their own standards and protocols to manage the network. To overcome the problem of trust among the nodes in a vehicular network, researchers have proposed many models for trust evaluation in VANETs. Considering the security of a vehicular network, trust is one of the most important challenges (*Hasrouny et al., 2017*; *Saini, Alelaiwi & Saddik, 2015*; *Yao et al., 2017*). Messages received by a node, such as information about accidents, traffic congestion, construction work in progress, natural disasters, and others, require the integrity of the information to be verified before action is taken. Vehicles with malicious intentions can divert traffic along specific routes. Many different techniques and methods have been adopted as approaches for trust evaluation over the years (*Yao et al., 2017*).

Over the years, several models for "vehicular network TM" have been proposed, and many researchers continue to work on TM models. Different techniques are being used to achieve the highest level of security using trust. As trust is a qualitative property, it is difficult to standardize. When the next generation of vehicular networks is implemented in the future, standardized TM would be necessary.

The purpose of our review is to conduct an in-depth exploration of the current models that have been presented to ensure that trust exists among the nodes of vehicular networks. Second, we aim to identify the strengths and weaknesses of existing models. Our final objective is to explore the potential implementation of context awareness in vehicular networks. The vehicle network is a dynamically changing network thus requires dynamic solutions. The dynamic system should be able to adapt according to the context to deal with the changes. Context awareness is one of the AI solutions that provides adaptivity to a system. Adaptivity is the transforming ability to fit in a certain condition, context awareness can enable adaptation (*Schmidt, Beigl & Gellersen, 1999*). As context awareness

is an emerging area of artificial intelligence (AI), context awareness has many advantages yet to be explored.

An adaptive mechanism needs to be used to ensure trust in a dynamic system (*Yan, Zhang & Vasilakos, 2016*). The importance of an environment-adaptive TM scheme for VANETs was discussed (*Pathan, 2016*) and it was concluded that the unwavering security infrastructure of VANETs lacks the ability to meet the security requirements. The trust evaluation system must be flexible and robust to adapt to different situations (*Saini, Alelaiwi & Saddik, 2015*). The TM deals with vast variable information, context awareness is the way to use available information depending on the situation, context awareness does not rely on static system inputs. In a research study (*Pathan, 2016*), authors discussed the need for VANET TM schemes to be adaptive to the environment. A study on the implications of AI in context awareness (*Kofod-Petersen & Cassens, 2005*) considered the usefulness of AI techniques to build the context. We conclude our study by presenting researchers with advice to enable them to solve different problems in vehicular networks.

## Trust

Trustworthiness is the level of legitimacy, and trust can be evaluated against the data or node in VANETs and ITS. Apart from securely delivering data between nodes, the integrity of the received content is important (*Ma, Wolfson & Lin, 2011*; *Yao et al., 2017*; *Zhou et al., 2015*). Trust is an important component of vehicular network security (*Ma, Wolfson & Lin, 2011*; *Sumithra & Vadivel, 2018*). Other researchers (*Ma, Wolfson & Lin, 2011*; *Saini, Alelaiwi & Saddik, 2015*; *Yao et al., 2017*) also discussed trust as a key security challenge owing to the dynamic nature of a vehicular network. Trust could be static where a trust level is assigned to certain types of vehicles/nodes such as RSUs or "police cars"; on the other hand, it could be dynamic where trust is developed based on interaction (*Ma, Wolfson & Lin, 2011*).

An objective of VANET security is to prevent messages from being forged during communication (*Hasrouny et al., 2017*). In repute-based systems, it is difficult for a vehicle that recently joined the network to prove its trustworthiness to the other nodes. In VANETs, the trustworthiness of data is of greater significance than that of a node because of the nature of these ad hoc networks (*Hasrouny et al., 2017*; *Raw, Kumar & Singh, 2013*). Data centric trust and verification in V2V communication in VANETs need to be studied in more detail. Trust evaluation techniques can be classified into three types (*Hasrouny et al., 2017*; *Raya & Hubaux, 2007*; *Saini, Alelaiwi & Saddik, 2015*; *Soleymani et al., 2017*; *Sumithra & Vadivel, 2018*; *Yao et al., 2017*) as shown in Fig. 2. The TM model that evaluates trust based on the available information is data centric. Entity centric TM model considers the trust associated with vehicle nodes.

## TM

The responsibility of the TM system is to manage the real-time and long-term trust of the nodes in a network. Second, to determine the legitimacy of the message received from different nodes in the network. Most of the early TM models were based on data and entity

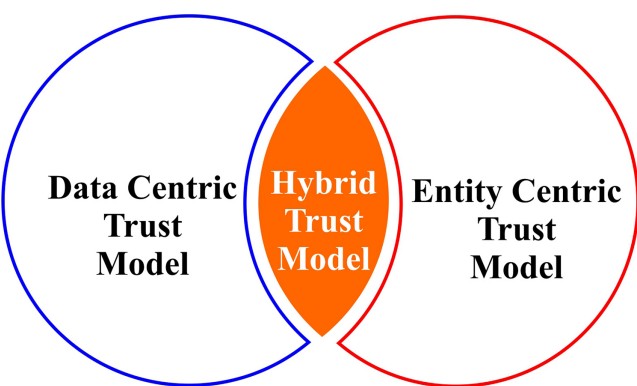

**Figure 2 Trust evaluation techniques classified into three types.**

now the trend is moving towards hybrid TMs. The main problem areas include trust evaluation during fewer node availability, uninterrupted centralized connectivity, and dealing with the dynamic nature of the network. Moreover, researchers are facing new challenges while developing TMs for IoV.

### Rationale for the review

IoV is an emerging field that lacks standards and needs to be standardized before IoV is implemented as part of smart cities. TM is one of the key components of IoV security. Previously, several TM models for VANET/ITS were presented, and these studies serve as the basis for IoV. In the absence of a comprehensive analytical study, this SLR provides a clear and comprehensive overview of the available evidence. In addition, our work helps to identify research gaps in our current understanding of the field. The objective of this review is to provide a basis for the developers of TM models for IoV. Our findings laid the foundation for the intelligent TM System.

### Intended audience

The article is intended to support academic and industry researchers working on TM in ad hoc vehicle networks, in particular the emerging field IoV. The work is not limited to IoV, the findings are equally beneficial for those working on IoT security. The open challenge sections are specifically presented to researchers who are new to the field of vehicle communication and need a strong research gap.

The remainder of the paper is organized as: Section 2 presents the methodology of the systematic literature review (SLR), Section 3 contains the results and discussion, and Section 4 discusses the findings and challenges.

## SURVEY METHODOLOGY

This research was conducted by using a method known as systemic literature review with the purpose of analyzing the limitations of this topic in a targeted way. Furthermore, we opted for meta-synthesis to help us identify problematic issues in current techniques and establish workable projections for further research in the area. The systematic study is based on existing PRISMA guidelines (*Kitchenham, 2004*).

**Table 1 The formulated protocol for this specific study based on the guidelines of a renowned review protocol PRISMA.**

| Title | Clear definition of problem with type of study |
| --- | --- |
| Abstract | Provide a comprehensive overview of the study being discussed, starting from the background. Review criteria, methodology, results, and key findings |
| Methodology | Objectives<br>Selection criteria<br>Scope of the study<br>Limitations |
| Introduction | In the sense of what is already understood, the rationale for the review. A clear statement of problems being addressed, comparison, outcome and study design |
| Results Discussion | Discussion on the outcomes with respect to research questions.<br>Limitations of the study<br>Open research challenges |
| Conclusions | Summary of the whole study conducted, general understanding of the results, and suggestions for future research |

## Justification of SLR

### 1. Lack of SLR in TM

As discussed in the "existing review" section, few review studies have focused on TM in VANETs; furthermore, an SLR has not yet been carried out for the IoV. There has been a surge in SLRs that collect and review existing knowledge, to help new researchers entering this particular field of research.

### 2. Lack of focused studies

Most of the review studies on VANETs encompass the entire security paradigm, with TM included as a single section. Contrary to this, our study focuses entirely on TM in vehicular networks.

### 3. Support related fields

This SLR would be equally helpful for those interested in IoT security, especially for device-to-device communication. Security has always been concerned with the IoT, many security threats emerge during IoT communication (*Mollah, Azad & Vasilakos, 2017*) and trust is one of the solutions to ensure security (*Jing et al., 2014*). Second, the SLR is also intended to serve to apply context awareness to different types of ad hoc networks.

## Review protocol

The review protocol for this study is based on the guidelines of a renowned review protocol PRISMA (*Moher et al., 2009*) and (*Kitchenham, 2004*). We formulated the following protocol for this specific study as in Table 1.

## Objectives of the study

Table 2 lists the research questions that cover the research objective of our SLR. The primary objective of this SLR is to analyze the core techniques that are used to manage trust among nodes in a vehicular network and motivated the construction of (RQ1): "What are the current techniques for trust evaluation/management in the

**Table 2 The research questions that cover the research objective of the SLR.**

| No. | Research question | Deliverable/Outcome |
|-----|-------------------|---------------------|
| 1 | What are the current techniques for trust evaluation/management in the vehicle network? | In-depth analysis of current trust evaluation and management techniques<br>Comprehend all the available models in a single study for further research |
| 2 | To what extent are the proposed models effective? | Weak and strong properties of current TM models considering real-time implementation<br>Identification of the gray areas of current models, that need to be investigated<br>Effectiveness analysis in term of trust evaluation<br>Open research challenges for researchers |
| 3 | Is context awareness suitable for the trust establishment in the vehicle network? | Contrast study of context awareness and vehicular network<br>Enlighten new research dimension for vehicular networks |

vehicular network?" To identify the unique and useful aspects of each model (RQ2): "What improvements are being proposed for solving these problems?" To determine the weaknesses and strengths of each study, we highlight the gray areas of current models, which are expected to open new research dimensions. Finally (RQ3) asks "Is context awareness suitable for the development of trust in a vehicular network?" We believe that the answers we provide to these questions culminate in a comprehensive study that not only offers new perspectives, but also opens new research dimensions.

## Selection criteria

### 1. First phase

This study focuses on the existence of trust among nodes in vehicular networks. To formulate the search terms, we used different literature surveys on vehicular networks and searched online repositories. We initially started with the terms listed in the following section. A total of 256 articles were collected at the end of the first phase.

### Key terms

Following key terms and search queries are searched in different repositories. The key search terms are driven by research questions using PRISMA guidelines.

- IoV
- VANET
- ITS
- Vehicular network
- Trust evaluation
- Trust management
- V2V communication
- MANET
- "VANET" or "IoV" or "ITS" and "trust management" or "trust evaluation"

- "VANET" or "IoV" and "trust management" and "V2V communication"
- "MANET" and "trust management" or "evaluation" and "context awareness"
- "IoV" and "trust management" and "V2V communication"
- "Internet of vehicles" and "trust evaluation" and "V2V communication"
- "Context aware" and "VANET" and "trust management" or "trust evaluation"

### 2. Second phase

Because of the large number of articles that were retrieved in first phase, the second step involved tool-based filtering. Filtering by using the citation tool allowed us to identify duplicate studies from different sources and versions. At the end of this phase, we had a collection of 120 articles from different repositories.

### 3. Third phase

In this phase, the focus was "abstracts" other than the entire metadata. By considering the abstract of each paper, irrelevant data could be removed, 73 papers were excluded by this process which resulted in 47 articles remaining in our collection.

### 4. Fourth phase

Finally, all the remaining papers were read in detail. This process led to the exclusion of papers that did not meet the selection criteria or standard or were beyond the scope of our work. During this phase 16 articles were excluded, finally, 31 research articles were selected to conduct the study. The SLR selection process is illustrated in Fig. 3.

## Scope of study

Our study focuses on the following statement:

In-depth analysis of present trust evaluation and management in vehicular networks and implications of context awareness in trust establishment in vehicular networks.

## Limitations of study

The following related areas are beyond the scope of our study:

- Routing protocols for the VANETs
- Cyber-attacks on VANETs

## RESULTS AND DISCUSSION

The intense, in-depth study we conducted yielded significant results. In this section, we summarize our findings, followed by a discussion. All three of the research questions were used to analyze the papers with respect to the findings. Before the research questions are discussed, selected statistics are presented below. Finally, following a detailed analytical discussion, we present the open research challenges and directions to researchers in the field.

## Type of studies

Table 3 presents the trends of studies conducted for trust evaluation and lists the year, category, and type of article ("J" for journal or "C" for conference). The studies were

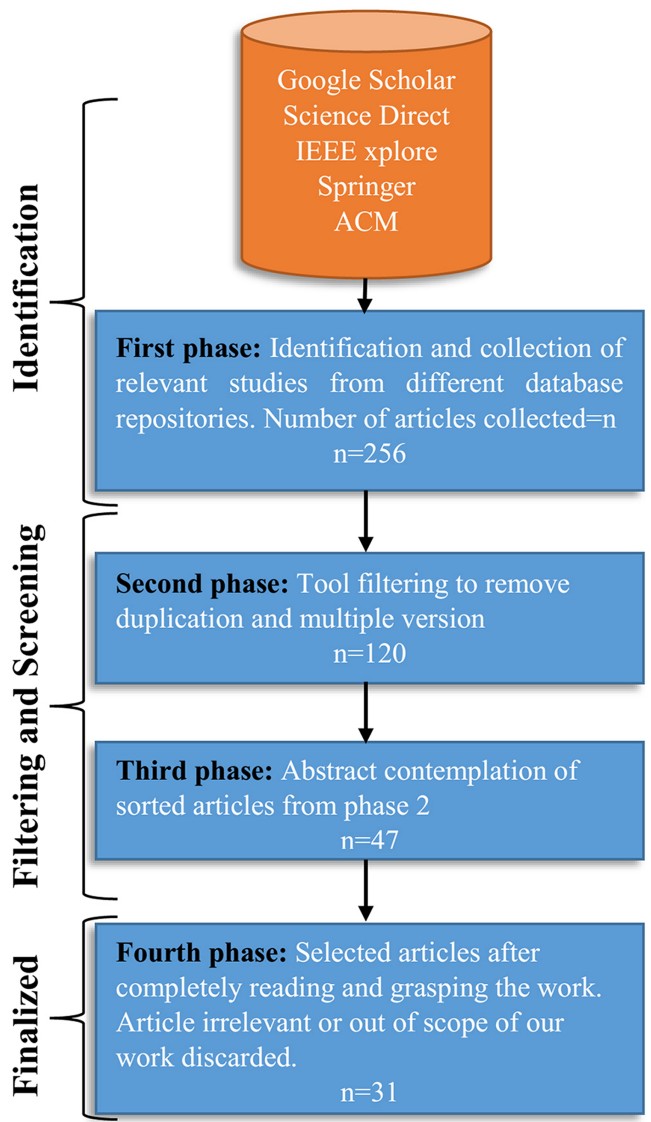

**Figure 3** The SLR selection process, and filtering in each phase.

categorized as data centric, entity centric, or hybrid. Our final collection of selected papers contained 13 "conference" articles and 18 "journal" articles. This number is based on the limitation of our study, and might vary for any other study. Figure 4 shows the number of publications on TM published in more than a decade. Figure 4 also shows the number of papers excluded during the fourth phase while filtering the abstracts.

## Critical characteristics of TM models

Table 4 presents certain important features of the TM model that were identified, that is, the use of a central authority (CA), the authentication that was used, new node

**Table 3 The trends of studies conducted for trust evaluation and lists the year.**

| Article | Year | Category | Journal/Conference |
|---|---|---|---|
| *Ahmad, Franqueira & Adnane (2018)* | 2018 | Hybrid | J |
| *Gazdar, Belghith & Abutair (2018)* | 2018 | Entity | J |
| *Soleymani et al. (2017)* | 2017 | Entity | J |
| *Ahmed, Al-Rubeaai & Tepe (2017)* | 2017 | Hybrid | J |
| *Biswas, Sanzgiri & Upadhyaya (2016)* | 2016 | Hybrid | C |
| *Hussain et al. (2016)* | 2016 | Hybrid | C |
| *Sedjelmaci & Senouci (2015)* | 2015 | Hybrid | J |
| *Haddadou, Rachedi & Ghamri-Doudane (2014)* | 2015 | Entity | J |
| *Rawat et al. (2015)* | 2015 | Data | J |
| *Ya, Shihui & Bin (2015)* | 2015 | Entity | J |
| *Rostamzadeh et al. (2015)* | 2015 | Hybrid | J |
| *Alagar & Wan (2015)* | 2015 | Entity | C |
| *Abdelaziz, Lagraa & Lakas (2014)* | 2014 | Entity | C |
| *Wahab, Otrok & Mourad (2014)* | 2014 | Hybrid | J |
| *Shaikh & Alzahrani (2014)* | 2014 | Data | J |
| *Wei, Yu & Boukerche (2014)* | 2014 | Hybrid | C |
| *Yang (2013)* | 2013 | Hybrid | J |
| *Chen & Wei (2013)* | 2013 | Hybrid | J |
| *Li et al. (2013)* | 2013 | Hybrid | C |
| *Zhang, Chen & Cohen (2013)* | 2013 | Hybrid | J |
| *Monir, Abdel-Hamid & El Aziz (2013)* | 2013 | Entity | C |
| *Rehman et al. (2013)* | 2013 | Data | C |
| *Li et al. (2012)* | 2012 | Entity | J |
| *Gazdar et al. (2012)* | 2012 | Hybrid | C |
| *Sahoo et al. (2012)* | 2012 | Entity | J |
| *Mármol & Pérez (2012)* | 2012 | Entity | J |
| *Minhas et al. (2011)* | 2011 | Entity | J |
| *Biswas, Misic & Misic (2011)* | 2011 | Entity | C |
| *Wu, Ma & Zhang (2011)* | 2011 | Data | C |
| *Chen et al. (2010)* | 2010 | Data | C |
| *Raya et al. (2008)* | 2008 | Data | C |

mechanism, uncertainty handling, and validation. These are a few of the most critical aspects of any TM system. A detailed discussion of the significance of these properties can be found in the discussion section.

## Simulation tool used in TMs

Most of the papers contain a discussion of the simulator that was used in the particular study. These simulators are listed in Table 5. The purpose of specifying the simulator is to lay the ground for researchers to choose simulation tools for further study.

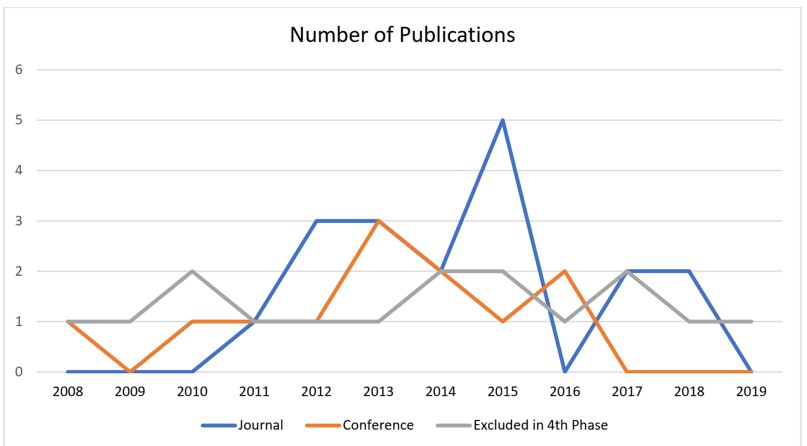

**Figure 4 The number of publications on trust management published in more than a decade.**

## Existing review studies

To the best of our knowledge, a few review articles on trust evaluation and management have been reported. A few studies partially considered trust, but these reviews were not sufficiently comprehensive. The review studies in which TM in vehicular networks was discussed are presented in Table 6 (*Arif et al., 2019*; *Kerrache et al., 2016*; *Soleymani et al., 2015*; *Sumithra & Vadivel, 2018*). Other than focused studies, several review articles on VANET security included a discussion on trust. This includes *Sumithra & Vadivel (2018)* who, in their review study presented at a conference, discussed the fundamentals of VANETs and related security. Their focus was on fuzzy logic-based systems. Another conference presentation (*Arif et al., 2019*) on a survey of trust relating to VANETs, focused on secure routing and the quality of service in future technical and non-technical directions. A journal review article (*Kerrache et al., 2016*) was published on TM in VANETs, including a security overview, attacks, and available trust solutions. Another review study on TM specifically selected ten TM models and their performance was analyzed (*Soleymani et al., 2015*), the second focus of this study was the use of Fuzzy logic in TM.

Other than the above-mentioned focused studies, several reviews (*Hasrouny et al., 2017*; *Raya & Hubaux, 2007*; *Saini, Alelaiwi & Saddik, 2015*) were conducted on ad hoc network security and VANETs. These broad studies paid slight attention to the significance of TM in VANETs.

**RQ1. What are the current techniques for trust evaluation/management in the vehicle network?**

### Approach by TMs

The core approaches adapted by TM models are listed in Table 7, in which models that have certain key approaches in common are combined. By analyzing the data provided in Table 7, it is easy to identify the most and least commonly adapted approaches and the most common aspects of different models. The commonly

**Table 4 The important features of the TM model that were identified.**

| Article | CA (Central Authority) | Authentication/PKI | New node mechanism | Uncertainty handling |
|---|---|---|---|---|
| Ahmad, Franqueira & Adnane (2018) | yes (for role based) | no | 0.5 initial value | no |
| Gazdar, Belghith & Abutair (2018) | no | no | no | no |
| Soleymani et al. (2017) | no | yes | no | no |
| Ahmed, Al-Rubeaai & Tepe (2017) | no | yes | no | no |
| Biswas, Sanzgiri & Upadhyaya (2016) | yes | yes | no | no |
| Hussain et al. (2016) | yes | yes | no | no |
| Sedjelmaci & Senouci (2015) | no | no | no | no |
| Haddadou, Rachedi & Ghamri-Doudane (2014) | no | RSA, elliptic curve cryptography | no | Markov chain |
| Rawat et al. (2015) | no | x | no | Probabilistic approach |
| Ya, Shihui & Bin (2015) | yes | yes | new node initial trust value 0.5 | By location |
| Rostamzadeh et al. (2015) | no | no | no | no |
| Alagar & Wan (2015) | yes, CA, GTA | yes | Start with 0 trust | no |
| Abdelaziz, Lagraa & Lakas (2014) | no | no | new node initial trust value 0.5 | no |
| Wahab, Otrok & Mourad (2014) | no | no | no | Dempster-Shafer |
| Shaikh & Alzahrani (2014) | no | no | no | no |
| Wei, Yu & Boukerche (2014) | no | no | no | no |
| Yang (2013) | no | no | 0.5 (Based on similar vehicle) | no |
| Chen & Wei (2013) | yes | yes | no | Dempster-Shafer evidence |
| Li et al. (2013) | yes, RMC | no | no | no |
| Zhang, Chen & Cohen (2013) | yes | no | no | no |
| Monir, Abdel-Hamid & El Aziz (2013) | yes, GTA | no | Start with 0 | By calculation |
| Rehman et al. (2013) | no | no | no | no |
| Li et al. (2012) | no | yes | no | no |
| Gazdar et al. (2012) | no | no | no | no |
| Sahoo et al. (2012) | no | no | All new vehicles are trusted | no |
| Mármol & Pérez (2012) | yes | no | no | no |
| Minhas et al. (2011) | no | no | no | no |
| Biswas, Misic & Misic (2011) | no | Local public key | no | no |
| Wu, Ma & Zhang (2011) | no | no | no | no |
| Chen et al. (2010) | no | no | no | no |
| Raya et al. (2008) | CA | PKI | no | no |

employed approaches were extracted from Table 7 and are plotted in Fig. 5 for easy understandability. The bar graph in Fig. 5 highlights the top ten parameters used in different studies. Experience is the topmost property adapted by most of the TM models, followed by neighbor opinion and vehicles with special roles.

Table 5 The simulators used by different TM models.

| Article | Simulator |
| --- | --- |
| *Ahmad, Franqueira & Adnane (2018)* | VEINS, SUMO, OMNET++ |
| *Gazdar, Belghith & Abutair (2018)* | not discussed |
| *Soleymani et al. (2017)* | NS2, SUMO |
| *Ahmed, Al-Rubeaai & Tepe (2017)* | OMNET++ discrete event, network simulator |
| *Biswas, Sanzgiri & Upadhyaya (2016)* | not discussed |
| *Hussain et al. (2016)* | not discussed |
| *Sedjelmaci & Senouci (2015)* | NS-3 |
| *Haddadou, Rachedi & Ghamri-Doudane (2014)* | NS-2, SUMO |
| *Rawat et al. (2015)* | not discussed |
| *Ya, Shihui & Bin (2015)* | not discussed |
| *Rostamzadeh et al. (2015)* | MATLAB |
| *Alagar & Wan (2015)* | not discussed |
| *Abdelaziz, Lagraa & Lakas (2014)* | NS-2 |
| *Wahab, Otrok & Mourad (2014)* | MATLAB, network simulator, MobiSim traffic |
| *Shaikh & Alzahrani (2014)* | SWANS++ |
| *Wei, Yu & Boukerche (2014)* | not discussed |
| *Yang (2013)* | not discussed |
| *Chen & Wei (2013)* | not discussed |
| *Li et al. (2013)* | NA |
| *Zhang, Chen & Cohen (2013)* | not discussed |
| *Monir, Abdel-Hamid & El Aziz (2013)* | MATLAB |
| *Rehman et al. (2013)* | not discussed |
| *Li et al. (2012)* | Groove Net |
| *Gazdar et al. (2012)* | Veins |
| *Sahoo et al. (2012)* | not discussed |
| *Mármol & Pérez (2012)* | TRMSim-V2V |
| *Minhas et al. (2011)* | SWANS |
| *Biswas, Misic & Misic (2011)* | not discussed |
| *Wu, Ma & Zhang (2011)* | NS-3 |
| *Chen et al. (2010)* | not discussed |
| *Raya et al. (2008)* | MATLAB |

Table 6 The review studies in which "vehicular network TM" is discussed.

| Review study | Year | Type of publication | Focus |
| --- | --- | --- | --- |
| *Sumithra & Vadivel (2018)* | 2018 | Conference | Fuzzy logic |
| *Gillani, Ullah & Niaz (2018)* | 2018 | Conference | Secure routing |
| *Kerrache et al. (2016)* | 2016 | Journal | Attacks |
| *Soleymani et al. (2015)* | 2015 | Journal | Fuzzy logic |

**Table 7 The core approaches adapted by TM models.**

| Article | Approach |
|---|---|
| *Ahmad, Franqueira & Adnane (2018)*, *Ahmed, Al-Rubeaai & Tepe (2017)*, *Chen et al. (2010)*, *Li et al. (2012)*, *Mármol & Pérez (2012)*, *Minhas et al. (2011)*, *Monir, Abdel-Hamid & El Aziz (2013)*, *Rostamzadeh et al. (2015)*, *Sahoo et al. (2012)*, *Soleymani et al. (2017)*, *Wei, Yu & Boukerche (2014)*, *Yang (2013)* | • Experience-based |
| *Ahmad, Franqueira & Adnane (2018)*, *Alagar & Wan (2015)*, *Chen et al. (2010)*, *Minhas et al. (2011)*, *Monir, Abdel-Hamid & El Aziz (2013)*, *Raya et al. (2008)*, *Zhang, Chen & Cohen (2013)* | • Role-based |
| *Abdelaziz, Lagraa & Lakas (2014)*, *Ahmed, Al-Rubeaai & Tepe (2017)*, *Haddadou, Rachedi & Ghamri-Doudane (2014)*, *Li et al. (2013)*, *Mármol & Pérez (2012)*, *Monir, Abdel-Hamid & El Aziz (2013)*, *Rostamzadeh et al. (2015)*, *Sahoo et al. (2012)*, *Sedjelmaci & Senouci (2015)* | • Neighbor opinion |
| *Ahmed, Al-Rubeaai & Tepe (2017)*, *Biswas, Sanzgiri & Upadhyaya (2016)*, *Chen & Wei (2013)*, *Gazdar, Belghith & Abutair (2018)*, *Mármol & Pérez (2012)*, *Wei, Yu & Boukerche (2014)*, *Yang (2013)* | • Direct and indirect experience |
| *Chen & Wei (2013)*, *Gazdar et al. (2012)*, *Rawat et al. (2015)*, *Raya et al. (2008)*, *Soleymani et al. (2017)*, *Wahab, Otrok & Mourad (2014)*, *Wei, Yu & Boukerche (2014)* | • Probability, Markov, Dempster-Shafer, Bayesian inference |
| *Alagar & Wan (2015)*, *Biswas, Misic & Misic (2011)*, *Biswas, Sanzgiri & Upadhyaya (2016)*, *Li et al. (2013)*, *Wu, Ma & Zhang (2011)* | • RSU managed trust |
| *Sahoo et al. (2012)*, *Sedjelmaci & Senouci (2015)*, *Wahab, Otrok & Mourad (2014)* | • Cluster-based |
| *Rostamzadeh et al. (2015)*, *Shaikh & Alzahrani (2014)*, *Soleymani et al. (2017)*, *Ya, Shihui & Bin (2015)* | • Location trust association |
| *Mármol & Pérez (2012)*, *Monir, Abdel-Hamid & El Aziz (2013)*, *Zhang, Chen & Cohen (2013)* | • Trust evaluated by central authority |
| *Minhas et al. (2011)*, *Sedjelmaci & Senouci (2015)* | • Majority Opinion |
| *Li et al. (2012)*, *Li et al. (2013)* | • Centralized trust management |
| *Gazdar, Belghith & Abutair (2018)*, *Haddadou, Rachedi & Ghamri-Doudane (2014)* | • Malicious node detection |
| *Ahmed, Al-Rubeaai & Tepe (2017)*, *Wei, Yu & Boukerche (2014)* | • Direct trust |
| *Shaikh & Alzahrani (2014)*, *Ya, Shihui & Bin (2015)* | • Time-based |
| *Alagar & Wan (2015)*, *Minhas et al. (2011)* | • Agent-based |
| *Soleymani et al. (2017)* | • Fuzzy logic<br>• Local ID authentication |
| *Ahmed, Al-Rubeaai & Tepe (2017)* | • Attack based model |
| *Ahmad, Franqueira & Adnane (2018)* | • Mobility behavior |
| *Biswas, Sanzgiri & Upadhyaya (2016)* | • long term trust relationship |
| *Hussain et al. (2016)* | • Person(driver) based<br>• Email-ID associated trust<br>• Social media |
| *Sedjelmaci & Senouci (2015)* | • Attack based |

| Table 7 (continued) | |
| --- | --- |
| **Article** | **Approach** |
| *Haddadou, Rachedi & Ghamri-Doudane (2014)* | • Credit allocation<br>• Distributed trust evaluation |
| *Rawat et al. (2015)* | • Deterministic |
| *Ya, Shihui & Bin (2015)* | • Response-time |
| *Abdelaziz, Lagraa & Lakas (2014)* | • Continuous link stability<br>• Piggybacking<br>• Local trust evaluation |
| *Wahab, Otrok & Mourad (2014)* | • Hope (1,2,3)<br>• Reward-based |
| *Wei, Yu & Boukerche (2014)* | • Recommendation |
| *Yang (2013)* | • Similarity (data, node) theory |
| *Chen & Wei (2013)* | • Beacon-based |
| *Li et al. (2013)* | • Reputation-based |
| *Zhang, Chen & Cohen (2013)* | • Peer-based opinion |
| *Monir, Abdel-Hamid & El Aziz (2013)* | • Driver ID based trust |
| *Rehman et al. (2013)* | • Thread-based |
| *Li et al. (2012)* | • Reputation score<br>• Feedback score(opinion) |
| *Gazdar et al. (2012)* | • Distributed trust management |
| *Sahoo et al. (2012)* | • Ant colony message routing |
| *Mármol & Pérez (2012)* | • Reward or punishment<br>• 3 level fuzzy trust |
| *Minhas et al. (2011)* | • Decentralized |
| *Biswas, Misic & Misic (2011)* | • ID-based<br>• Certificate less<br>• Public key |
| *Wu, Ma & Zhang (2011)* | • Observation reports<br>• Feedback |
| *Raya et al. (2008)* | • Different trust combined<br>• Weighted voting |

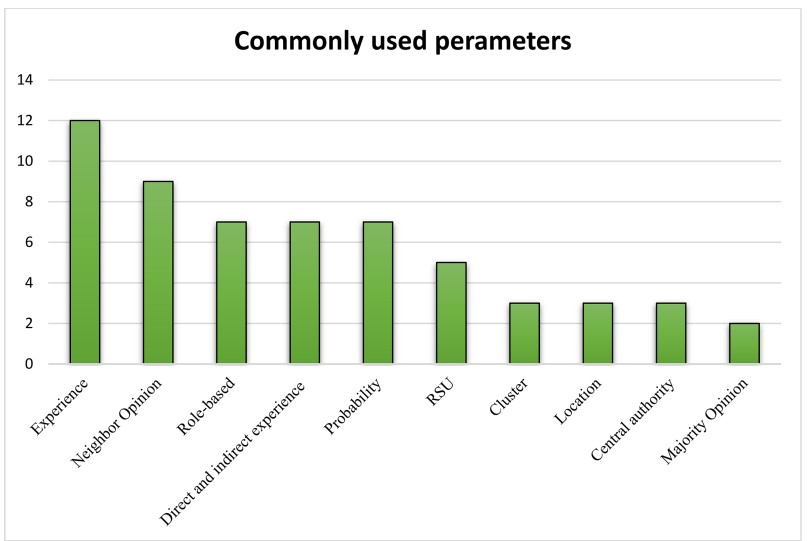

**Figure 5** The top ten parameters used in different studies.

The core techniques, methodologies, and related components used by each TM model are summarized in Table 8. As discussed in "Introduction", TM models can be categories as: data centric, entity centric, and hybrid. Previously, researchers mostly focused on data centric TMs, after which they realized that node properties are an important part of any TM, resulting in the proposal of many entity centric models. Hybrid models are a combination of both of these techniques. Other than these three categories, certain other characteristics play an important role in TM, as discussed in this section. The aim of the discussion is to clarify the concepts behind different techniques that are expected to lay the foundation for unification during the standardization process. Furthermore, the discussion highlights certain aspects warranting further research.

**1. Data centric TM models**

These models are based on data (message) for trust evaluation. A data centric model (*Chen et al., 2010*), with the aim of improving the scalability using trust, was proposed. The authors evaluated the trust by using an opinion-based approach to identify malicious nodes by evaluating the trust. In addition, they used a cluster and role-based approach with the majority opinion being one of the key elements of this approach.

A few similar models (*Minhas et al., 2010b*; *Raya & Hubaux, 2007*), in which a neighboring vehicle provides its opinion on the trustworthiness of events, were presented. The trust is computed on the basis of the highest vote. A drawback of these models is that messages can only be communicated among specified nodes in one group. These models can be used for distributed attacks using "neighbor node opinion." A second drawback of these models is that opinion calculation methods are not clearly defined.

Other researchers (*Shaikh & Alzahrani, 2014*) presented a decentralized TM system for VANETs. The scheme is based on different values calculated for the location and time to evaluate the trust. Their model seems effective in certain situations, but has limitations. Based on similarity, another TM model (*Yang, 2013*) used the similarity

**Table 8 The summary of core techniques, methodologies, and related components used by each TM.**

| Article | Methodology |
| --- | --- |
| *Ahmad, Franqueira & Adnane (2018)* | Trust over data and vehicle is established and finally, message trust is evaluated "Role-based vehicles" and experience play an important role |
| *Gazdar, Belghith & Abutair (2018)* | Neighboring-node opinion is considered. Negative trust introduced in this model. Probability is used for uncertainty handling |
| *Soleymani et al. (2017)* | Trust determined by fuzzy logic, experience, and probability. Local ID authentication. Experience value is saved in the node. Location verification using time and distance. The sender location is a key factor in this model |
| *Ahmed, Al-Rubeaai & Tepe (2017)* | Neighbor opinion, consistency, and similarity-based trust evaluation. Three subcategories of trust: events, nodes, and recommendations |
| *Biswas, Sanzgiri & Upadhyaya (2016)* | RSU add and manage trust. Long term trust relationship is maintained. RSU supported by CA at the top layer. The model assumes that all the vehicles in the networks registered with the CA and maintains a certain level of trust |
| *Hussain et al. (2016)* | Based on email and social trust. Trust is managed by a CA. Trust is associated with user |
| *Sedjelmaci & Senouci (2015)* | Three categories of trust (trusted, suspicious and attacker). Majority voting-based trust. The cluster head is responsible for trust management |
| *Haddadou, Rachedi & Ghamri-Doudane (2014)* | Credit-based model removes malicious nodes. Selfish nodes get participation rewards. RSA cryptographic scheme used |
| *Rawat et al. (2015)* | The trust evaluated by calculating distances using signal-strength, arrival-time and location. The probability of a malicious node is calculated by Bayesian |
| *Ya, Shihui & Bin (2015)* | A time frame-based trust protocol. Feedback on report from neighboring nodes. Selfish and malicious node identification. To validate the trust response time is used |
| *Rostamzadeh et al. (2015)* | The TM model is subdivided in multiple modules. Three security levels to assure trust. Trust is allocated to road segments |
| *Alagar & Wan (2015)* | Role-based vehicles act as agents. RSU assesses the context environment and manages the system |
| *Wahab, Otrok & Mourad (2014)* | Role-based vehicles act as agents. RSU assesses the context environment and manages the system |
| *Shaikh & Alzahrani (2014)* | Neighbor vehicles chose the most reliable nodes among adjacent nodes to be used for hopping. Local trust management by each vehicle. Piggybacking is used by the last message |
| *Wei, Yu & Boukerche (2014)* | Cluster-based TM model. Misbehaving vehicles detected by speed variation. Reward allowed for motivation. Dempster-Shafer theory for counter verification. Some nodes act as watchdog in the system |
| *Yang (2013)* | A decentralized trust management system. The scheme is based on change in the location and time used to calculate the trust |
| *Chen & Wei (2013)* | The Bayesian rule method is used to calculate the trust level. Dempster-Shafer is used for handling uncertainty |
| *Li et al. (2013)* | Nodes monitor neighboring nodes. Trust is based on opinion. Final trust is calculated from new and previous data. RSU is the part of trust evaluation |
| *Zhang, Chen & Cohen (2013)* | Before forwarding received messages, peer inputs their opinion. Role-based nodes are also part of the model. The central authority manages the whole network |
| *Monir, Abdel-Hamid & El Aziz (2013)* | Association of driver ID with trust. CA and RSU are responsible for opinion gathering and trust evaluation. A malicious node penalizing system introduced |
| *Rehman et al. (2013)* | During the hopping process trust level is increased. The multiple chains of trust threads created to finally evaluate trust |
| *Li et al. (2012)* | The trust is observed as a reputation score of any node network. Experience is involved in building a reputation. Feedback is recorded against the reporting vehicle. The centralized authority shall serve as controlling authority |
| *Gazdar et al. (2012)* | Based on vehicle behavior using a Markov chain. All vehicles monitor their surroundings. Local trust management |
| *Sahoo et al. (2012)* | An ant colony-based cluster TM model. The cluster is made using different parameters such as direction, speed, and others. Higher trust level and neighbor's opinion decide cluster head |

*(Continued)*

| Table 8 (continued) | |
| --- | --- |
| **Article** | **Methodology** |
| *Mármol & Pérez (2012)* | Prior experience, neighbor suggestion are the main trust evaluators in this model with three levels of local trust storage. CA managed node trust. The reward and punishment system. The trust is determined in a fuzzy system |
| *Minhas et al. (2011)* | An intelligent agent-based approach. Trust based on experience, role, priority and majority opinion |
| *Biswas, Misic & Misic (2011)* | The model is based on node ID. Nodes are identified by the digital signatures. Public key verification is used for authentication |
| *Wu, Ma & Zhang (2011)* | An RSU based TM model. The RSU is responsible for trust evaluation. Feedback process adds up to trustworthiness |
| *Chen et al. (2010)* | A role-based TM model. Experience and neighbor node opinion |
| *Raya et al. (2008)* | The distributed TM model is based on role-based desperation. Government-related vehicles hold a higher trust level. Bayesian inference and Dempster-Shafer theory are used to handle uncertainty. Event-specific trust is observed |

between a node and the data to evaluate the trust level. Although a unique method of similarity was introduced, the technique was not validated. The trust was calculated based on direct and indirect experiences. The experimental details were not discussed, although the model seems to belong to the research field theoretically.

A model based on vehicle behavior and a Markov chain (*Gazdar et al., 2012*) was presented. The Markovian state transition was used to allocate and evaluate the trust of a node. In this model, all the vehicles act as monitors within their related perimeter. The model is distributed and maintains the record of trust locally. In another study (*Wu, Ma & Zhang, 2011*). the researchers proposed an RSU-based data centric TM model. An earlier data centric TM model (*Raya et al., 2008*) considers role-based desperation, where nodes related to governmental organizations hold more trust credibility. In contrast, the model also considers the node trust level. Bayesian inference and the Dempster-Shafer theory are used to handle uncertainty.

The data centric approach is used by earlier TMs, which lack the use of information associated with the node (entity). Entity centric and hybrid models have better capabilities than data centric models. It can be concluded from the above discussion that, even though the data centric approach is less effective, yet this approach is important for TM. It can play a key role in a hybrid TM system (*Yao et al., 2017*).

**2. Entity centric TM models**

These models are based on building trust against vehicle (entity). In a recently presented entity centric TM model (*Soleymani et al., 2017*) the level of trust is assessed by fuzzy logic, using experience and probability. Authentication is based on the local ID and the location of the sender also plays a major role in the model. Another entity centric distributed TM model (*Haddadou, Rachedi & Ghamri-Doudane, 2014*) is based on the allocation of credit to the nodes used for TM. The model identifies the malicious node and motivates selfish nodes by awarding credits if they participate. The node is initially assigned a credit value, which increases or decreases over the time period.

A trust protocol based on a time frame (*Ya, Shihui & Bin, 2015*) is presented with the aim of identifying malicious vehicles in the network. The trust value of a node is

determined by the feedback from the neighboring nodes. A hybrid TM model that considered location privacy (*Chen & Wei, 2013*) was presented. The presented technique is based on beaconing, where the movement of nodes plays an important role. A combination of direct and indirect trust was used in this model to evaluate the trust, which was validated using Dempster-Shafer evidence. An ant colony-based cluster TM model was proposed (*Sahoo et al., 2012*). Clusters were formed by using different parameters such as direction and speed. The head of a cluster was selected based on a higher trust level. Trust was based on neighbor opinion.

A model based on prior experience, neighbor-suggestions, and CA was used to build node trust (*Mármol & Pérez, 2012*), which was achieved by using the reward and punishment method. The trust was determined in a fuzzy fashion to facilitate assessment by the system. Other entity centric models (*Cui et al., 2019*; *Li et al., 2013*) used a mix of CA and cryptography. The primary problem with these models is that, once the vehicle has been authenticated, it cannot act maliciously. A decentralized agent-based TM model (*Minhas et al., 2010a*) using role, experience, and opinion, was presented. Another TM model (*Biswas, Misic & Misic, 2011*) was based on node-ID and local authentication by the public key, where digital signatures play a key role in this model for node identification. The assumption that "an authenticated node is trustable" is not realistic in VANETs. A scheme to ensure trust during communication in a VANET was proposed (*Li et al., 2012*). They set the "trust" as the reputation score of any node in the network. A message from highly reputed vehicles is considered trusted. However, experience is involved in building a reputation. Another unique aspect of their work is the feedback against the reporting vehicle.

An entity centered framework for trust evaluation without a centralized system was proposed (*Gazdar, Belghith & Abutair, 2018*). The framework focuses on immediate knowledge between adjacent cars. Their system is also effective for fake message detection. The solution is based on the self-organization of nodes in a network rather than depending on centralized authorities. The trust level of a vehicle is evaluated using different parameters. Neighboring vehicles play an important role in trust measurement. Based on the discussion and analysis of entity centered techniques, it is firmly established that these models are the core of any TM model. Without considering the entity, the performance of TM models is compromised. On the other hand, simply relying on the entity for trust evaluation is also not the ultimate solution.

By discussion and analysis of entity centric techniques, it is firmly established that these models are the core of any TM models. Without considering entity the performance of TM models is compromised. On the other hand, relying just on the entity for trust evaluation is also not the ultimate solution.

### 3. Hybrid TM models

Most of the recent models are hybrid models, combining data and evaluate the legitimacy of the received message. A proposed hybrid trust approach (*Yao et al., 2017*) for VANETs is based on the assignment of classified weights to different entities of the system. The model applies experience and utility theory, and the authors assumed certain

attributes and the pre-assignment of weight. A renowned group (*Wei, Yu & Boukerche, 2014*) applied a hybrid approach, based on a combination of Bayesian rules to find the trust and Dempster-Shafer theory for handling uncertainty. The approach is based on direct and indirect trust to calculate the overall trust. A hybrid framework for TM, based on neighbor nodes, was presented (*Ahmed, Al-Rubeaai & Tepe, 2017*). The model divides trust into three categories: events, nodes, and recommendations. Based on consistency and similarity, the trust for a certain event is calculated and updated. The presence of legitimate events and nodes in the system is an assumption. In a long-term trust relationship model (*Biswas, Sanzgiri & Upadhyaya, 2016*) the main trust evaluation body is the RSU supported by CA at the top. The model assumes that all the vehicles in the networks are registered with the CA and maintain a certain level of trust.

A trust evaluation study based on cluster and intrusion detection is presented to deal with different attacks (*Sedjelmaci & Senouci, 2015*). The trust level is evaluated using majority voting by neighbor nodes. The model is subdivided into three: local, cluster head, and RSU. Trust is managed by the cluster head. The node is categorized in the list of trusted, suspicious, and attackers. *Rawat et al. (2015)* worked on a combination of two different approaches, probabilistic and deterministic. The probability of a malicious node is calculated using Bayesian. The deterministic part calculates the trust message by calculating distances using signal strength, arrival time, and location. By most reliable nodes among adjacent cars (*Abdelaziz, Lagraa & Lakas, 2014*), a new idea of filtering is used as a forwarding process. The implementation of the system avoids engaging in a network with cars recognized as likely dishonest users. A trust list is maintained locally by every vehicle, which is based on node interaction. Node velocity is used to determine the neighbors. The last message received serves as an integral part in evaluating trust since piggybacking is used.

In a cluster-based trust framework, misbehaving vehicles are sorted by assessing their speed variation (*Wahab, Otrok & Mourad, 2014*). The system is based on a reward scheme for motivation. To counter the trust and untrusted nodes, the Dempster–Shafer module is set in place as a watchdog. The cluster head plays the main role in their model. The model seems very complicated and unrealistic in real situations. In a study (*Li et al., 2013*) based on a centralized TM, all the nodes monitor neighboring nodes and forward their opinion to the centralized system. The central system finally calculates the trust value based on new and previous data. RSU also plays an important role in this system.

In a study (*Zhang, Chen & Cohen, 2013*) researchers worked on a peer opinion based TM model, and peers after receiving a message provided their opinion and forwarded the message. The model combines many sub-modules, such as propagation, trust evaluation, and peer-to-peer trust modules. Finally, trust is monitored and managed by a CA. An RSU-managed TM model, using role and experience, was presented (*Monir, Abdel-Hamid & El Aziz, 2013*) and is unlike other models that associate the driver ID with trust evaluation. The trust level of a vehicle increases with time in terms of experience. A penalty for malicious nodes is also part of the system, which reduces the trust level of the node in the future.

Overall, hybrid models have more adaptability and dynamicity. As vehicular networks require dynamic solutions, it is essential to design a TM model in a hybrid manner. The hybrid models allow multiple dimensions and the properties of both data and nodes to be used for trust evaluation.

**RQ2. To what extent are the proposed models effective?**

In this section, we discuss the critical aspects of a TM model with respect to existing models. The purpose of the discussion is to facilitate the establishment of the effectiveness of the current models.

### Assumptions made by TM models

Most models made certain assumptions about their TM processes that create inflexibility. In a dynamic environment, assumptions decrease the importance of the proposed solution. For instance, assumptions made by *Chen et al. (2010)* and *Raya et al. (2008)*, that special-vehicles are present in every situation, are unrealistic. Another assumption made by *Sedjelmaci & Senouci (2015)*, "RSU is always available" which is practically unfeasible. Likewise, *Abdelaziz, Lagraa & Lakas (2014)* assume that all nodes are moving with the same velocity, which does not match the nature of the network. So much so, every model has some assumptions that make it hard to implement in real-time. The assumptions made by the TM models are listed in Table 9. However, assumptions regarding the implementation process prove to be a major hurdle. During a real traffic event, assuming the presence of special vehicles, RSU availability, pre-assigned trust, and many others can create serious problems while evaluating the trust. During a random event and network scenario, a TM model must not assume any variable to be completely implementable.

### Centralized or decentralized vehicular network

There has always been a divide on whether ad hoc networks should be centralized or decentralized. Certain models (*Abdelaziz, Lagraa & Lakas, 2014*; *Gazdar et al., 2012*; *Haddadou, Rachedi & Ghamri-Doudane, 2014*; *Saini, Alelaiwi & Saddik, 2015*; *Shaikh & Alzahrani, 2014*) considered decentralization as their approach, whereas others applied a centralized approach (*Hussain et al., 2016*; *Li et al., 2012*, *2013*; *Mármol & Pérez, 2012*; *Monir, Abdel-Hamid & El Aziz, 2013*). In a recent study (*Arif et al., 2019*) the authors recommended centralized validation for TM. The main issue is the ultimate connectivity, which is difficult to achieve in these networks. In addition, for security, centralized data storage and management is essential. Although a practical solution would be to periodically synchronize trust data with a centralized management system, completely relying on centralized management is not the best practice. On the other hand, having a solely decentralized system causes a serious threat to the ad hoc networks.

### Importance of authentication

The first step in any security system is authentication with a vehicular network. Here, the point of dispute is "If the authentication alone fulfills the security requirements." A VANET is likely to experience a security breach after the legitimate approval of an

**Table 9 The assumptions made by the TM models.**

| Article | Assumptions made by the studies |
|---|---|
| *Ahmad, Franqueira & Adnane (2018)* | Government vehicles, public transport and vehicles with a higher mileage are highly trusted |
| *Ahmed, Al-Rubeaai & Tepe (2017)* | At least one true (legitimate) event and node must be the part of network |
| *Biswas, Sanzgiri & Upadhyaya (2016)* | Predefined RSU trust value<br>CA always keeps a trust record of all vehicles |
| *Sedjelmaci & Senouci (2015)* | Uninterrupted RSU availability<br>All RSUs are connected to the wire |
| *Haddadou, Rachedi & Ghamri-Doudane (2014)* | All types of nodes (legitimate, malicious and selfish) are present in the system<br>All nodes are allocated with Initial credit |
| *Rawat et al. (2015)* | Presence of at least one malicious vehicle in the network |
| *Ya, Shihui & Bin (2015)* | The omnipresence of reliable intersection vehicle<br>Presence of "geographic in-charge" nodes in the network<br>Use of uniform technology by all the vehicle |
| *Rostamzadeh et al. (2015)* | Initial trust allocation against road segment |
| *Alagar & Wan (2015)* | OBU is always a secure device and uniform all over the network |
| *Abdelaziz, Lagraa & Lakas (2014)* | The neighbor nodes have the same velocity |
| *Shaikh & Alzahrani (2014)* | The event always occurs at the end of the road segment<br>Nodes are in line of sight |
| *Chen & Wei (2013)* | Frequent single hopping message broadcast only<br>Third party CA is always trusted<br>Pseudonyms change frequently |
| *Zhang, Chen & Cohen (2013)* | Pre-allocated roles |
| *Li et al. (2012)* | Keys are managed by OBU<br>The clock is in OBU and always secure |
| *Gazdar et al. (2012)* | All vehicles in the network must report the event<br>All vehicles must forward receive messages |
| *Sahoo et al. (2012)* | Highway traffic only |
| *Mármol & Pérez (2012)* | Same route daily |
| *Minhas et al. (2011)* | Vehicle manufacturers issue security certificates |
| *Chen et al. (2010)*, *Raya et al. (2008)* | Special vehicles are always present in network |
| *Biswas, Misic & Misic (2011)*, *Gazdar, Belghith & Abutair (2018)*, *Hussain et al. (2016)*, *Li et al. (2013)*, *Monir, Abdel-Hamid & El Aziz (2013)*, *Rehman et al. (2013)*, *Soleymani et al. (2017)*, *Wahab, Otrok & Mourad (2014)*, *Wei, Yu & Boukerche (2014)*, *Wu, Ma & Zhang (2011)*, *Yang (2013)* | Not discussed |

authenticated node. In certain studies, authenticated users were assumed to always broadcast legitimate messages (*Cui et al., 2019*; *Li et al., 2013*; *Mármol & Pérez, 2012*). In a real situation, this may result in highly adverse security consequences, such as Sybil, DOS and DDOS, node-impersonation attacks and, message-tempering (*Kerrache et al., 2016*; *Saini, Alelaiwi & Saddik, 2015*; *Sakiz & Sen, 2017*). Furthermore, *Sakiz & Sen (2017)* explain in their work that how an already authenticated node can be compromised by an attacker for blackhole and wormhole attacks. An authenticated node can be used for session hijacking, explain *Hasrouny et al. (2017)*.

Authentication must be dynamic and efficient in order to meet the requirements of IoT/IoV (*Challa et al., 2020*), for which researchers are working on lightweight authentication schemes (*Vasudev et al., 2020*). *Wazid et al. (2020)* proposed a lightweight authentication scheme for IoT sensor data. Another research work (*Wazid et al., 2018*) on lightweight user authentication for internet of drones for direct communication is proposed. Blockchains can be a useful tool to build lightweight authentication solutions (*Jangirala, Das & Vasilakos, 2019*). The result shows that lightweight authentication schemes are suitable and secure for dynamic networks. Even though authentication is an essential aspect of an ad hoc network, we cannot rely solely on authentication.

### New node entering the network

A critical issue during TM that is not taken into consideration is the entry of new nodes into the network. The question "how to assign a new node with initial trust" remains unanswered. Certain studies (*Abdelaziz, Lagraa & Lakas, 2014*; *Ahmad, Franqueira & Adnane, 2018*) assigned a new node a trust value of 1 or 0.5, but the reason for this approach is not discussed in any of these papers. An intense study needs to be conducted to formalize the mechanism whereby a new node is introduced to a vehicular network.

### Cluster approach to handle trust

A number of studies (*Chen et al., 2010*; *Sahoo et al., 2012*; *Sedjelmaci & Senouci, 2015*; *Wahab, Otrok & Mourad, 2014*) adopted a cluster approach for managing trust. It is observed in most of the studies that a cluster creates inflexibility in a network. A VANET has a dynamically changing topology (*Xu et al., 2020*), and inflexibility can cause the network to malfunction. Nonetheless, this does not decrease the importance of clustering in certain scenarios. In conclusion, the cluster approach can be used in TM but in limited scenarios. Cluster management needs to be defined and standardized to be used successfully. Cluster head selection has also been discussed in certain studies.

### Role-based approaches

The presence of special vehicles (vehicles from governmental organizations such as ambulances, police vehicles, or fire engines) in the network requires a role-based approach (*Ahmad, Franqueira & Adnane, 2018*; *Alagar & Wan, 2015*; *Haddadou, Rachedi & Ghamri-Doudane, 2014*; *Monir, Abdel-Hamid & El Aziz, 2013*; *Zhang, Chen & Cohen, 2013*). Role-based nodes are considered trustworthy in these models and are assigned a higher trust level than other nodes. The question that remains unanswered is the approach to follow when special vehicles are present in every event. Practically, it is not possible to have special vehicles everywhere; eventually, the subtraction of role-based vehicles from the network at any time would affect the performance of the TM model. Role-based nodes and their use have been the topic of many studies, and this approach might be useful in certain situations.

### RSU-based approaches

In the early phases of VANET research, RSUs became an essential part of the VANET architecture (*Behravesh & Butler, 2016*). Models (*Alagar & Wan, 2015*; *Biswas, Misic & Misic, 2011*; *Biswas, Sanzgiri & Upadhyaya, 2016*; *Li et al., 2013*; *Monir, Abdel-Hamid & El Aziz, 2013*; *Sedjelmaci & Senouci, 2015*; *Wu, Ma & Zhang, 2011*) that used the RSU as the main authority for TM were proposed. A "trust value" was assigned to an RSU depending on its location and coverage (*Biswas, Sanzgiri & Upadhyaya, 2016*). Another RSU-based study (*Alagar & Wan, 2015*) presented a trust scheme using role-based vehicles. Further, they considered an RSU as a monitoring and managing node that also has a trust value, and the overall network was controlled by the CA. A TM model involving RSU-based scheme (*Wu, Ma & Zhang, 2011*) was proposed, in which an RSU receiving environmental change reports by all vehicles after receiving reports from the RSU is responsible for deciding whether any message is real or false. The feedback process serves to confirm the trustworthiness. However, relying on an RSU is not a practical approach, as RSUs are stationary nodes and RSU availability cannot be assured across the entire network (*Chi et al., 2013*; *Haydari & Yilmaz, 2018*). Despite all of these facts, the RSU continues to remain the main component of the VANET architecture and can be used wisely in certain situations or as a supporting node.

### Fuzzy logic in TM

Fuzzy logic is a very useful tool and is used in many AI solutions. Two research groups (*Mármol & Pérez, 2012*; *Soleymani et al., 2017*) used fuzzy logic in their work to evaluate trust. According to the first group (*Soleymani et al., 2017*) reason for using fuzzy logic is the uncertainty within VANETs. Their results show that fuzzy logic can be used positively in trust evaluation. In a review study (*Sumithra & Vadivel, 2018*) authors elaborated on the potential need for a fuzzy logic method for the TM model in VANETs. The study found coherence between VANET and fuzzy based solutions. The authors further mentioned that fuzzy approaches lack robustness and verity adoption, however, fuzzy methods seem to be a good solution if they are used in TM. The only disadvantage of fuzzy based TMs is the time consumed during the process of computing the trust because a vehicular network cannot afford time delays.

### Uncertainty handling while evaluating trust

A point of uncertainty always exists during trust evaluation (*Yao et al., 2017*).
The majority of models are unable to discuss the uncertainty in their solutions. A TM model without the ability to handle uncertainty is considered an incomplete solution. Models that considered uncertainty while evaluating trust (*Chen & Wei, 2013*; *Haddadou, Rachedi & Ghamri-Doudane, 2014*; *Monir, Abdel-Hamid & El Aziz, 2013*; *Wahab, Otrok & Mourad, 2014*; *Wei, Yu & Boukerche, 2014*; *Ya, Shihui & Bin, 2015*), processed the uncertainty by using probability approaches such as Bayesian, the Dempster Shafer theory, and Markov processes. Researchers designing a TM model need to prioritize uncertainty handling.

### Use social media

Social media could form a new addition to TM in IoV networks. TM in social vehicular networks was discussed on the basis of email and social media-based trust (*Hussain et al., 2016*). The model relies on CA to manage trust. The idea of considering social media was presented at a conference and seemed interesting, but is quite exposed to social media threats. Here it is important to note that the trust is associated with the user rather than the node. Driver-based trust faces the problem of disassociation, which might lead to miscalculation of the trust value. However, social media is an essential part of next-generation design (*Zia, Shafi & Farooq, 2020*).

**RQ3 Is context awareness suitable for the development of trust in the vehicle network?**

### Context awareness

Vehicular networks are the most complex wireless networks and thus require flexibility in all aspects. Context awareness is an AI method that can be used to introduce flexibility to any system; contextual awareness is the ability to assess and adapt the environment (*Ramos, Augusto & Shapiro, 2008*). Very few studies in this field have been reported, and a few studies have partly considered the context in vehicular networks (*Ahmad, Franqueira & Adnane, 2018*; *Alagar & Wan, 2015*; *Rostamzadeh et al., 2015*). Our efforts to find the true essence of context awareness in these studies were unsuccessful. However, in related fields, such as distributed, ubiquitous, and mobile ad hoc networks, several studies related to context awareness have been reported.

A study of IoT architecture for smart cities (*Gaur et al., 2015*) used context awareness for scenario building. Their work focuses on the representation of knowledge of the smart city architecture through contextual awareness, which has become a necessity for ubiquitous systems (*Taconet & Kazi-Aoul, 2010*). Another study on interoperability in ubiquitous systems (*Strang & Linnhoff-Popien, 2003*) applied context awareness with positive results. A study on VANET/ITS (*Al-Sultan, Al-Bayatti & Zedan, 2013*) used context awareness to assess driver behavior. The outcome of the study was that context awareness has significant potential in relation to vehicular networks. The importance of context awareness in MANETs (*Moloney & Weber, 2005*) elaborated on the importance of context implementation in a system that is equipped with sensors. Moreover, the study suggested that any systems with sensor objects that can make good use of context awareness would be beneficial to vehicular networks while using node-to-node communication. Node-to-node context applications in MANETs were investigated (*Wibisono, Zaslavsky & Ling, 2010*) for the purpose of context building in MANETs. Context-aware systems are able to adapt to changes and increase the usability of any system and the efficacy thereof was discussed (*Maneechai & Kamolphiwong, 2014*) in relation to distributed systems. In a futuristic study on different aspects of vehicular network security (*Wan et al., 2014*), the use of context awareness in such systems is strongly recommended, especially with respect to security and safety. Context awareness for automatic TM is an important field of research (*Yan, Zhang & Vasilakos, 2014*).

A survey study (*Yürür et al., 2016*) on context awareness for the future of mobile systems, determined that a context-aware system has the ability to sense its environment. Furthermore, by using context the network increases its security, quality, and credibility. Their work concludes by highlighting the use of context for mobile-based systems. Furthermore, IoT architecture must be intelligent, and for that, context awareness is an effective method (*Sarkar et al., 2014*). The forthcoming networks are intelligent networks, for which context awareness is the key method to be adopted (*Bilen & Canberk, 2015*). A study on ambient computing (*Babu & Sivakumar, 2014*) discussed the ability of context awareness to enable adaptation for ubiquitous systems. Further, the researchers pointed out that a system that assesses the situation and acts accordingly must be equipped with context awareness. The challenge is to design context-based security middleware for a heterogeneous system. An earlier study determined that context awareness has special characteristics that support intelligent vehicle systems (*Sørensen et al., 2004*), but little work has been carried out in this direction. Further, they discuss the context that can be useful in V2V communication for intelligent vehicles.

Context awareness is a necessity for upcoming ad hoc networks, especially VANETs (*Aguilar, Jerez & Rodríguez, 2018*). Their work on context modeling has significant potential in wireless networks. Further, they indicated that the application of context to ad hoc networks may be advantageous. Context awareness was employed to enhance the security of wireless networks (*Lin, Yan & Fu, 2019*). This study was based on heterogeneous networks to show the implications of context awareness. Their positive results showed that applying context awareness to security increased the adaptivity.

In a VANET context-aware study for detecting misbehaving nodes (*Chi et al., 2013*), the authors applied context using neighboring nodes and used mobility information to construct the context. Their work yielded positive results that were the outcome of context awareness. A context knowledge representation scheme was presented (*Ruta et al., 2018*). Their work shows the implications of context in wireless networks, especially IoT-related technologies and VANETs, which strongly support our research question. In a recent study, context was employed to design a safety system in VANETs (*Shen et al., 2016*). The concept is for vehicles to share the road environment while driving safely. Routing can be improved by context awareness in wireless networks, especially in the IoT (*Dhumane et al., 2018*). A context-based VANET privacy model (*Emara, Woerndl & Schlichter, 2015*) that implemented contextual privacy successfully improved location tracing. The work was evaluated against the quality of service and found to be satisfactory even after the application of context awareness.

Based on the above discussion and relevant studies, it is established that context awareness has the potential to be applied to vehicular networks for TM. Nonetheless, the term context could be interpreted to have a general meaning and many related challenges would need to be overcome. As the context has already been considered for MANETs, it could be used as the foundation for VANET/IoV as well. A comparative overview is presented in Table 10 based on a published report (*Sharma & Kaushik, 2019*).

**Table 10 A comparative overview of VANET and MANET.**

| Property | VANET | MANET |
|---|---|---|
| Mobility | High | Low |
| Size | Large | Small |
| Movement | Geographic paths | Random in limited space |
| Protocol | 802.11p | 802.11a |
| Topology | Fast-changing | Slow changing |
| Power | Unlimited | Limited |
| Node speed | Very high | Low |
| Interaction time | Low | High |
| No of nodes | Unknown | Can be restricted |
| Anonymity | High | Low |
| Node variation | Very High | Low |

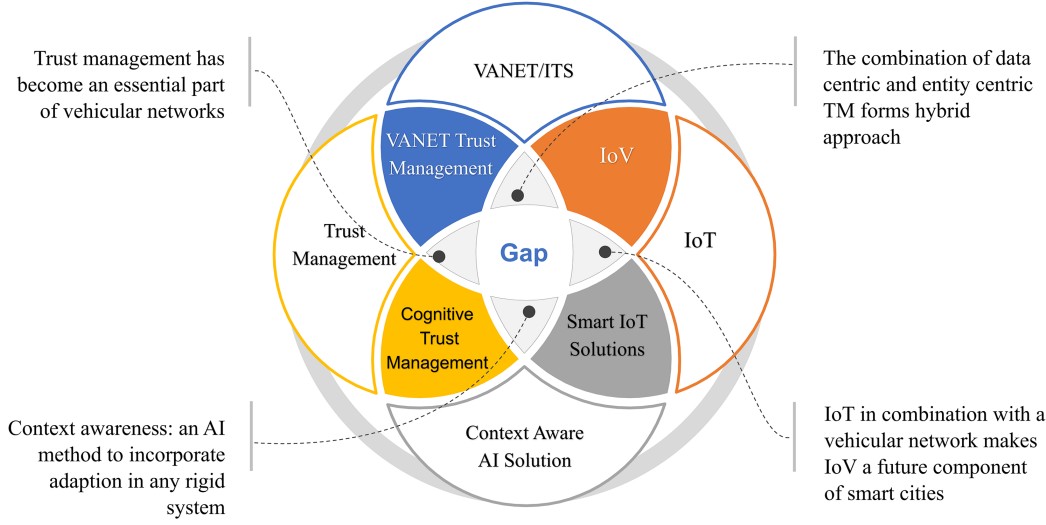

**Figure 6 An overview of the different technologies, the extent to which they overlap, related issues, and the potential future of these technologies.**

## Summary at a glance

An overview of the different technologies, the extent to which they overlap, related issues, and the potential future of these technologies is presented in Fig. 6. Furthermore, Table 11 summarizes the major properties of the TM models. Only journal articles were considered, the reason for the exclusion of the conference papers at this point is to be precise, second, they are unable to provide detailed information on methodology, experimental results and analysis. The aim of the summary is to produce a comprehensive yet brief description for researchers in the field. According to the statistics in Table 11, two very important aspects of the TM model have been ignored by most of the researchers i.e., hopping and beaconing. Most malicious activities occur during hopping and beaconing (*Jang et al., 2018*; *Suman, Srinivas & Rao, 2017*).

**Table 11 Summarizes the major properties of the TM models (only journal articles were considered).**

| Study | CA | Authentication | RSU | Experience | Swarm | Cluster | Opinion | Hopping | Special nodes | Fuzzy logic | Decentralized | Beaconing | Credit allocation | Node Initialization | Uncertainty |
|---|---|---|---|---|---|---|---|---|---|---|---|---|---|---|---|
| Ahmad, Franqueira & Adnane (2018) | yes | no | no | yes | no | no | no | no | yes | no | no | no | no | yes | no |
| Gazdar, Belghith & Abutair (2018) | no | no | no | yes | no | no | no | no | no | no | no | no | no | no | no |
| Soleymani et al. (2017) | no | yes | no | yes | no | no | no | no | no | yes | no | no | no | no | no |
| Ahmed, Al-Rubeaai & Tepe (2017) | no | yes | no | yes | no | no | yes | no | no | no | no | no | no | no | no |
| Wahab, Otrok & Mourad (2014) | no | no | no | no | no | no | no | no | no | no | no | no | yes | no | yes |
| Shaikh & Alzahrani (2014) | no | no | no | no | no | no | no | no | no | no | yes | no | no | no | no |
| Yang (2013) | no | no | no | yes | no | no | no | no | no | no | no | no | no | yes | no |
| Chen & Wei (2013) | yes | yes | no | no | no | no | yes | yes | no | no | no | no | no | no | yes |
| Zhang, Chen & Cohen (2013) | yes | no | no | no | no | no | yes | no | yes | no | no | no | no | no | no |
| Li et al. (2012) | no | yes | no | yes | no | no | yes | no | no | no | no | no | no | no | no |
| Sahoo et al. (2012) | no | no | no | yes | yes | yes | yes | no | no | no | no | no | no | yes | no |
| Mármol & Pérez (2012) | yes | no | no | yes | no | no | yes | no | no | yes | no | no | yes | no | no |
| Minhas et al. (2011) | no | no | no | yes | no | no | yes | no | no | no | yes | no | no | no | no |

**Table 12 Research challenges and directions related to gray areas.**

| S.No. | Related domain | Challenge | Research direction |
|---|---|---|---|
| 1 | TM frameworks | Development of intelligent TM | Trust management requires intelligent solutions, intelligent solutions in trust management are missing |
| 2 | Network | Choosing between a centralized or decentralized network | A comparative study is required between centralized and decentralized approaches for ad hoc networks |
| 3 | Intelligent computing | Implementing concepts of nature-inspired computing (NIC) in vehicular ad hoc networks | The vehicular network has similarities with nature-inspired computing such as swarm computing and ant colony. The scientific coherence needs to be explored |
| 4 | Network initialization | Introducing a new node in ad hoc network | While managing the trust new node entry in a network is a problem that needs to be addressed |
| 5 | Network cluster | Cluster management | The cluster formatting and management for vehicular network needs to be standardized |
| 6 | Uncertainty | Handling uncertainty while managing trust | One of the complex problems while working on trust evaluation is uncertainty. Comprehensive solutions are required to handle uncertainty |
| 7 | Social media | Use of social media for security | The role of social media in enhancing the vehicular network security needs to be explored |
| 8 | Cybersecurity | Preventing cyber attacks | The greatest threat to IoV security is cyber-attacks. IoV desperately requires cybersecurity protocols. |
| 9 | Artificial intelligence | Use of context awareness in ad hoc networks | The use of context awareness in ad hoc network security is a huge research area, that needs to be explored |
| 10 | Context awareness | Implementation and knowledge representing | Implementing context awareness is a huge research area, especially knowledge representation in ad hoc networks. An intense in-depth research exercise is required in this direction |

RSUs, an important component of the VANET infrastructure, were not considered in any of the studies. However, RSUs were included in most of the conference idea presentations. As an RSU is an important entity with many functionalities, it can be used to take maximum advantage of the resources.

Analyzing the statistics in Table 11 shows that TM is a significant problem. Most studies have ignored certain major aspects. Therefore, dynamic TM models that consider maximum aspects to attain the highest level of trust are in demand. The inclusion of all aspects in a single TM model can be challenging, therefore requires an adaptive solution. Context awareness is a method that provides flexible adaptation to a complex system (*Ramos, Augusto & Shapiro, 2008*).

## Open challenges and future research directions

Based on the analysis and observations of this study, the following are open research challenges that need to be addressed. This is expected to provide research directions for researchers interested in this field, Table 12 provides research challenges and directions related to identified gray areas. Furthermore, the future vision of IoV is discussed to provide clear goals, helping researchers to work on a broader canvas.

1. Designing intelligent TM frameworks for vehicular networks. From results, it can be inferred that TM is a complex problem thus requires intelligent systems. This will enhance the overall performance of the TM.

2. Choosing between a centralized or decentralized approach has always been debatable. A comparative analysis of centralized and decentralized approaches for VANETs is required. This analysis will provide clear benefits and drawbacks to each approach.

3. Nature-inspired computing has similarities with ad hoc vehicular networks. Detailed analytical study on the coherence of swarm computing with vehicular networks is required. Swarm computing, with its intelligent capabilities, has some characteristics similar to the vehicle network. Some research work involving ant-colony is conducted in this field.

4. Solving the "node initialization" problem for TM. Whenever a new node joins an ad hoc network different initial trust credits need to be allocated. The trust value allocation to a new node in a vehicular network is one of the challenging problems for TM models.

5. Cluster management for vehicular network TM models. The studies using cluster have no common approach for managing the trust. An in-depth study is required to analyze and provide standards. A cluster approach can also be used as a subpart of the network security solution.

6. Overcoming uncertainty while evaluating trust in VANETs. Uncertainty is one of the complex problems in trust evaluation, comprehensive solutions are required in this area to handle uncertainty. A few works have been carried out in this aspect of TM models.

7. Use of social media for security in vehicular networks. Social media has become a part of our lives, this research area needs further exploration for security implementation. Some works have been carried out in this area showing positive results.

8. Designing IoV cyber security protocols. IoV is exposed to cyber-attacks due to internet connectivity. It, therefore, requires standard security protocols, specially designed for IoV cyber-security.

9. Using context awareness for security in ad hoc networks. Context awareness provides intelligent solutions, existing work in this field shows positive results. Coherence between both fields needs to be further investigated.

10. Context knowledge representation methods for vehicular networks. Knowledge representation is one of the complex problems faced while applying context awareness. Before implementing the context awareness research needs to work out the knowledge representation methods.

11. Context awareness in the vehicular environment. Within this area some research work is carried out which shows promising results, further exploration is needed for better solutions.

## Framing the future vision of IoV, and recommendations

IoV is expected to become part of the future smart city and ITS, the vision of future IoV is set out in the following key points and provides researches on a broad-spectrum expectation.

1. Soon, IoV is expected to be fully or partially implemented as part of smart cities.

2. Conventional security is expected to be replaced with intelligent security solutions for vehicular networks.

3. Trust between nodes is expected to be a standard part of future vehicular network security, there is still a potential gap in this direction.

4. Fog and edge technology are also making way for vehicular networks that will be a good addition to the IoV.

5. Vehicular communication generates huge data in seconds, the Big Data has a huge potential in vehicular networks, yet to be explored. Driver behavior, traffic patterns, accident prevention, smooth traffic flow, and many other interesting discoveries and solutions can be made using Big Data analysis.

6. Context awareness has an enormous potential in the vehicular network yet to be explored and implemented in the future.

7. A mix of a centralized and decentralized network is also expected to be established.

8. Another aspect that is worthy of further investigation is the management of cloud resources during vehicular communication. Various problems arise with regard to the nature of the communication.

9. The social network seems to be part of IoV. Apart from the human social network, the social network of vehicles is also likely to be developed.

## CONCLUSION

The IoV is a potential part of the intelligent transport solution for the next wave of smart cities. As the IoV is still in the development phase, applicable standards need to be developed and recognized worldwide. Thus, the TM models in the IoV are yet to be standardized and transformed from VANETs and the ITS. We devoted our efforts to produce a systematic review in which we examined all available TM models. The outcomes of the meta-analysis highlighted all dimensions, including gray areas. Further, to fulfill the need for intelligent solutions, we tried to establish the potential benefits of context awareness in vehicular networks, which would not only facilitate intelligent TM but also address other security issues. Our work could be useful to develop standards for TM in vehicular networks. The results of the study are equally helpful for designing IoT security solutions, as device-to-device communication is a key aspect of the future IoT.

In terms of future research, we raised open questions for researchers in the Open Challenges section. Answering each question is likely to require intense research efforts. The outcome of our work is expected to be equally fruitful for other IoT solutions, ad hoc networks, and trust-related studies.

### Funding

The work is funded by Universiti Teknologi PETRONAS, Malaysia, YUTP-FRG grant number 0153AA-E90 and Universiti Teknologi PETRONAS, Malaysia GA Scheme. There

was no additional external funding received for this study. The funders had no role in study design, data collection and analysis, decision to publish, or preparation of the manuscript.

### Grant Disclosures

The following grant information was disclosed by the authors:
Universiti Teknologi PETRONAS, Malaysia, YUTP-FRG: 0153AA-E90.
Universiti Teknologi PETRONAS, Malaysia GA Scheme.

### Competing Interests

The authors declare that they have no competing interests.

### Author Contributions

- Abdul Rehman conceived and designed the experiments, performed the experiments, analyzed the data, prepared figures and/or tables, authored or reviewed drafts of the paper, and approved the final draft.
- Mohd Fadzil Hassan conceived and designed the experiments, analyzed the data, prepared figures and/or tables, authored or reviewed drafts of the paper, and approved the final draft.
- Kwang Hooi Yew analyzed the data, authored or reviewed drafts of the paper, and approved the final draft.
- Irving Paputungan analyzed the data, authored or reviewed drafts of the paper, and approved the final draft.
- Duc Chung Tran analyzed the data, authored or reviewed drafts of the paper, and approved the final draft.

### Data Availability

    This is a literature review article; there is no raw data.

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
