# Peer review of "State-of-the-art IoV trust management a meta-synthesis systematic literature review (SLR)"

_PeerJ Computer Science, doi:10.7717/peerj-cs.334_

## Round 0.1 · original submission · Major Revisions

The reviewers have identified a number of studies that have not been covered in the SLR so far. Since it is the goal of an SLR to provide a systematic approach to find and discuss literature, this means that the actual methodology (search terms, etc.) needs to be revised.

Apart from this, the reviewers have also identified a number of minor aspects that need to be addressed.

Reviewer 1 ·

Basic reporting

Revision:
-The section Open research issues should be enhanced(long term vision on future research challenges. This is very important for a good survey).
-Important references are missing and should be added:
-Mohammad Wazid,et al:LAM-CIoT: Lightweight authentication mechanism in cloud-based IoT environment. J. Netw. Comput. Appl. 150 (2020)
-Jun Zhou,et al:Secure and privacy preserving protocol for cloud-based vehicular DTNs.IEEE Transactions on Information Forensics and Security 10 (6), 1299-1314,2015.
-Mohammad Wazid, et al:Design and Analysis of Secure Lightweight Remote User Authentication and Key Agreement Scheme in Internet of Drones Deployment. IEEE Internet of Things Journal 6(2): 3572-3584 (2019).
-Harsha Vasudev,et al:Secure message propagation protocols for IoVs communication components. Comput. Electr. Eng. 82: 106555 (2020).
-Sravani Challa, et al: Design and analysis of authenticated key agreement scheme in cloud-assisted cyber-physical systems. Future Gener. Comput. Syst. 108: 1267-1286 (2020)

Experimental design

no comment

Validity of the findings

no comment

Additional comments

Revision:
-The section Open research issues should be enhanced(long term vision on future research challenges. This is very important for a good survey).
-Important references are missing and should be added:
-Mohammad Wazid,et al:LAM-CIoT: Lightweight authentication mechanism in cloud-based IoT environment. J. Netw. Comput. Appl. 150 (2020)
-Jun Zhou,et al:Secure and privacy preserving protocol for cloud-based vehicular DTNs.IEEE Transactions on Information Forensics and Security 10 (6), 1299-1314,2015.
-Mohammad Wazid, et al:Design and Analysis of Secure Lightweight Remote User Authentication and Key Agreement Scheme in Internet of Drones Deployment. IEEE Internet of Things Journal 6(2): 3572-3584 (2019).
-Harsha Vasudev,et al:Secure message propagation protocols for IoVs communication components. Comput. Electr. Eng. 82: 106555 (2020).
-Sravani Challa, et al: Design and analysis of authenticated key agreement scheme in cloud-assisted cyber-physical systems. Future Gener. Comput. Syst. 108: 1267-1286 (2020)

Reviewer 2 ·

Basic reporting

no comment

Experimental design

no comment

Validity of the findings

no comment

Additional comments

The authors conducted a systematic literature review (SLR) using PRISMA guidelines to find historical and current trends of trust management methodologies in IoV/VANETs. They have screened several conference and journal papers and selected 31 papers finally to provide an in-depth analysis of them. Authors lay out some open challenges and future research directions with various kinds of important summary tables. The following are some of my observations about the article.

1. Why context-awareness should be part of the review was not clear to me. It should be elaborated for the reader in the introduction part.

2. What exactly is a context in terms of TM should be defined?

3. Texts in 120-123 appear to be misplaced.

4. A table highlighting the gray areas that relate to the open research challenges will be very useful.

5. Keywords that are used in lines 220-225 show formatting error.

6. What exactly was the reason conference papers excluded in some places of the summary should be detailed?

7. How many journals and conference papers were excluded based on their filtering, might be important for the readers especially in the 3rd phase.

8. Please check the sentence of 280-282. Not clear.

9. Figure 4 can represent at least the fourth phase.

10. Assumptions section from line 429 should be elaborated more to relate to the table.

11. "In a real situation, this may result in highly adverse security consequences" such as?

12. Give references for 488-489.

13. Can table 3 have the information from fig 6 to show the technology overlap?

---

## Round 0.2 · Minor Revisions

Please make sure that you take into account the further papers mentioned by Reviewer 1.

Reviewer 1 ·

Basic reporting

This study focuses on trust management (TM) in the IoV/VANETs/ITS
(intelligent transport system). Trust has always been important in vehicular networks to ensure safety. A variety of techniques for TM and evaluation have been proposed over the years, yet few comprehensive studies that lay the foundation for the development of a “standard” for TM in IoV have been reported.

Important references are missing and should be added:
Sravani Challa, et al: Design and analysis of authenticated key agreement scheme in cloud-assisted cyber-physical systems. Future Gener. Comput. Syst. 108: 1267-1286 (2020)
Mohammad Wazid, et al:Design and Analysis of Secure Lightweight Remote User Authentication and Key Agreement Scheme in Internet of Drones Deployment. IEEE Internet of Things Journal 6(2): 3572-3584 (2019).
S Jangirala, et al:Designing secure lightweight blockchain-enabled RFID-based authentication protocol for supply chains in 5G mobile edge computing environment.IEEE Transactions on Industrial Informatics, DOI: 10.1109/TII.2019.2942389,2020
MB Mollah, et al:Secure data sharing and searching at the edge of cloud-assisted internet of things.IEEE Cloud Computing 4 (1), 34-42,2017.
Harsha Vasudev, et al:Secure message propagation protocols for IoVs communication components. Comput. Electr. Eng. 82: 106555 (2020)
Z.Yan,et al:A security and trust framework for virtualized networks and software‐defined networking.Security and communication networks 9 (16), 3059-3069,2016
Q.Jing,et al:Security of the Internet of Things: perspectives and challenges.
Wireless Networks 20 (8), 2481-2501, 2014
Z.Yan,et al: A survey on trust management for Internet of Things.
Journal of network and computer applications 42, 120-134,2014

Experimental design

no comment

Validity of the findings

no comment

Additional comments

Important references are missing and should be added:
Sravani Challa, et al: Design and analysis of authenticated key agreement scheme in cloud-assisted cyber-physical systems. Future Gener. Comput. Syst. 108: 1267-1286 (2020)
Mohammad Wazid, et al:Design and Analysis of Secure Lightweight Remote User Authentication and Key Agreement Scheme in Internet of Drones Deployment. IEEE Internet of Things Journal 6(2): 3572-3584 (2019).
S Jangirala, et al:Designing secure lightweight blockchain-enabled RFID-based authentication protocol for supply chains in 5G mobile edge computing environment.IEEE Transactions on Industrial Informatics, DOI: 10.1109/TII.2019.2942389,2020
MB Mollah, et al:Secure data sharing and searching at the edge of cloud-assisted internet of things.IEEE Cloud Computing 4 (1), 34-42,2017.
Harsha Vasudev, et al:Secure message propagation protocols for IoVs communication components. Comput. Electr. Eng. 82: 106555 (2020)
Z.Yan,et al:A security and trust framework for virtualized networks and software‐defined networking.Security and communication networks 9 (16), 3059-3069,2016
Q.Jing,et al:Security of the Internet of Things: perspectives and challenges.
Wireless Networks 20 (8), 2481-2501, 2014
Z.Yan,et al: A survey on trust management for Internet of Things.
Journal of network and computer applications 42, 120-134,2014

Reviewer 2 ·

Basic reporting

No comments

Experimental design

No comments

Validity of the findings

No comments

Additional comments

The authors have addressed most of my concerns and it improved the quality of the paper. Hence, I recommend accepting the paper for publishing in this journal.

---

## Round 0.3 · Minor Revisions

The paper is practically accepted, i.e., it will not be given to the reviewers again, since their comments have been addressed. However, I (the editor) will read it again.

Please take into account the following comments:
* In the abstract, it is claimed that 265 papers have been checked, but in the actual paper, it's 256.
* The tables are unfortunately not usable, since the numbers of the publications are different from numbering in the references. It is an absolute must that the publications are identifiable!
* In Table 12, there are 11 entries, while the according text features only 10. Is there a specific reason for this?

While the language is in general fine, please take into account the following comments:
* Please introduce an acronym before using it (e.g., "AI")
* If you have introduced an acronym, use it always. Do not switch between acronym and full form again and again (e.g. "TM(s)")
* While the paper has been proof-read by a professional service, there are some uncommon sentence structures. Most likely, there is just a word missing in each sentence, so please check this thoroughly in Lines 123-125, 149f, 166f, 405-406, 417-418, 421f, 442-446,
* There are some missing blanks here and there, e.g., Line 143, 618, and a lot of the table captions
* Line 192: "in ensure security" => "to ensure security"
* Line 304: number of Figure is missing
* Both "Dempster Shafer" and "Dempster-Shafer" are used within the paper. Please unify this. Same for "big data" and "Big Data"
* Line 380: Instead of saying "against fake message detection", you most likely mean "for fake message detection"?
* Line 407: "into three" => into three what?
* Line 415: "serves an integral" => "serves as an integral"
* Line 483: "0,1" => "0.1"
* Line 572: "for that context, awareness" => "for that, context awareness"
* Line 572: "next ear" => I don't actually get this...

Also, please provide all figures as vector graphics, since this increases readability and scalability of the figures.

Last but not least, please check the references thoroughly w.r.t. layout and content. For instance, the actual journal is missing for MANY entries, e.g., [17]. Make sure that you provide full information. Also, please check the author names for Athanasios Vasilakos, e.g., [26], [61]. I don't know why some additional given names have been added here. For [96], make sure that you add the full information about the patent.

---

## Round 0.4 · accepted · Accept

When preparing the camera-ready version, please take into account the following comments:
* Line 78: "to disaster" => "to a disaster"
* Line 120: Is there a special reason that "Context" is written here in uppercase mode? If not, please use lowercase here.
* Line 172: "article was" => "article is"
* Lines 207ff: Why is RQ1 not put in quotation marks?
* Line 225: Incomplete sentence
* Both "IoV" and "IOV" are used within the manuscript. This should be unified
* Lines 235: Please use bullet points; else, it's difficult to assess where a particular search term ends
* Lines 270: While it is claimed that 31 studies are discussed in detail, these include 13 conference papers and 19 journal papers (i.e., 32 papers overall). This doesn't sum up.
* Line 321: "models, are" => "models are"
* Both "entity-centric" and "entity centric" are used within this paper. This should be unified (also in Figure 2 and Figure 6)
* Lines 436-437: Incomplete sentence
* Lines 458-460: Incomplete sentence
* Line 462: "assumes that" => "assume that"
* Line 474: "author" => "the authors"
* Line 489: "explains" => "explain"; "that how" => "how"
* Line 491: "explains" => "explain"
* Line 504: Year is missing for Abdelaziz et al.
* Line 548: "author" => "the authors"
* Line 549/560/645/653/676/728 (and most likely at other places throughout the paper - please search and replace!): "trust model" => "TM" (in general, please make sure that you use the acronyms once they have been introduced)
* Line 550: "The author" => "The authors"
* Both "fuzzy-based" and "fuzzy based" are used within the paper. Please unify this.
* Acronym DST is never introduced
* Line 643: ")." => "."
* Line 664/704/Table 12: If there is no special reason to write "Intelligent", please use "intelligent"
* Line 667: "vehicular ad hoc networks" => "VANETs"
* Line 700: "intelligent transport systems" => "ITSs"
* Please check the references thoroughly. It seems as if there were some issues with the compilation of the literature section, e.g., why is lowercase lettering used for many journal or conference names; "Patents G" is not an editor; Jing et al. is missing the journal name; if DOIs are named, they should be named everywhere (or nowhere); Citeseer is not a publisher; conference names should be abbreviated everywhere or nowhere; Suman et al. is missing the journal name and further information; Engineering E is not an author; either use "Proceedings of" everywhere or nowhere
* Why is Figure 4 referring to decades (also in the running text), but actually showing years?
* Table 3: Entry (Yang 2013): "hybrid" => "Hybrid"
* Table 3/7: "Markova" => "Markov"
* Both "neighbor opinion" and "neighbor-opinion" are used. This should be unified.
* Table 8: "TM model" => "TM"
* Both "role based" and "role-based" are used. This should be unified.
* Table 10: "low" => "Low" (2x)
* Table 11: "decentralized" => "Decentralized"
* Both "context awareness" (which is correct) and "context-awareness" are used. This should be unified.